# RoboHiMan: A Hierarchical Evaluation Paradigm for Compositional Generalization in Long-Horizon Manipulation

## Abstract

Enabling robots to flexibly schedule and compose learned skills for novel long-horizon manipulation under diverse perturbations remains a core challenge. Early explorations with end-to-end VLA models show limited success, as these models struggle to generalize beyond the training distribution. Hierarchical approaches, where high-level planners generate subgoals for low-level policies, bring certain improvements but still suffer under complex perturbations, revealing limited capability in skill composition. However, existing benchmarks primarily emphasize task completion in long-horizon settings, offering little insight into compositional generalization, robustness, and the interplay between planning and execution. To systematically investigate these gaps, we propose **RoboHiMan**, a hierarchical evaluation paradigm for compositional generalization in long-horizon manipulation. RoboHiMan introduces **HiMan-Bench**, a benchmark of atomic and compositional tasks under diverse perturbations, supported by a multi-level training dataset for analyzing progressive data scaling, and proposes **three evaluation paradigms** (vanilla, decoupled, coupled) that probe the necessity of skill composition and reveal bottlenecks in hierarchical architectures. Experiments highlight clear capability gaps across representative models and architectures, pointing to directions for advancing models better suited to real-world long-horizon manipulation tasks. Anonymous project website: https://robohiman.github.io/.

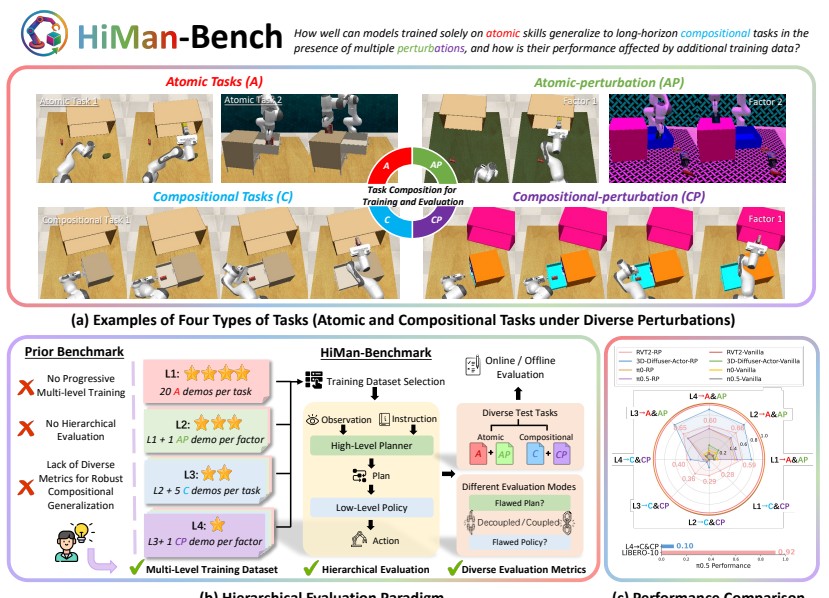

Figure 1: **RoboHiMan Overview.** To evaluate compositional generalization, RoboHiMan introduces: **(a)** HiMan-Bench with four task types: atomic (A), atomic-perturbation (AP), compositional (C), and compositional-perturbation (CP); **(b)** a hierarchical evaluation paradigm with diverse metrics and progressive training data (L1–L4), where L1 uses minimal atomic data and L4 provides larger datasets; **(c)** Extensive experiments highlight critical performance gaps across training datasets and evaluation modes, often overlooked by prior benchmarks (notation "X → Y" denoting training on Level X and evaluation on task category Y).

## 1 INTRODUCTION

In the field of robot manipulation, a long-term goal is to enable robots to perform diverse long-horizon tasks (Zhang et al., 2024; Shi et al., 2025; Chen et al., 2024; 2025b;e;a). However, achieving this goal requires overcoming a fundamental challenge: *compositional generalization*. Specifically, we expect robots to, much like humans, master a set of atomic skills (e.g., opening a drawer, picking up objects) through imitation learning, and flexibly schedule and compose them to complete new long-horizon tasks (e.g., opening a drawer then placing an object inside) (Chen et al., 2025f). However, in real-world applications, compositional generalization intensifies as robots must contend with various perturbations, such as changes in lighting, object appearance, or camera poses (Pumacay et al., 2024). Therefore, evaluating compositional generalization involves testing whether models can effectively compose skills under such diverse conditions. To systematically study this problem, we focus on a central research question: *How well can models trained solely on atomic skills generalize to long-horizon compositional tasks in the presence of various perturbations?*

Existing manipulation benchmarks (James et al., 2020; Liu et al., 2023; Chen et al., 2025c; Mees et al., 2022; Zhang et al., 2024; Chen et al., 2025f; Han et al., 2025) have played an important role in advancing the field by providing diverse long-horizon tasks for training and evaluation. However, they exhibit notable limitations: most benchmarks focus on evaluating models on complete long-horizon tasks without systematically examining the flexible composition of atomic skills (James et al., 2020; Liu et al., 2023; Chen et al., 2025c). DeCoBench (Chen et al., 2025f) considers skill composition but lacks an in-depth analysis of how environmental perturbations affect compositionality. Colosseum (Pumacay et al., 2024) only assesses the robustness of atomic skills under perturbations but does not evaluate multi-stage compositional tasks. More critically, existing benchmarks make it difficult to disentangle whether failures arise from insufficient planning, poor execution, or sensitivity to environmental perturbations (Mees et al., 2022; Zhang et al., 2024; Chen et al., 2025f; Han et al., 2025). The rough metric and task design of them leave open questions about which module is responsible for failures, which hinders the development of a new method in this domain.

To address these limitations, we propose **RoboHiMan**, A a hierarchical evaluation paradigm for compositional generalization in long-horizon manipulation, which makes two core contributions. The first is **HiMan-Bench**, a new benchmark dedicated to compositional generalization in robot manipulation. Building on the design principles of DeCoBench and Colosseum, HiMan-Bench evaluates whether models can compose atomic skills to accomplish long-horizon tasks under diverse environmental perturbations, such as changes in object appearance, size, lighting, and distractors. Compared with prior benchmarks, HiMan-Bench explicitly measures the effect of perturbations on skill composition (advancing beyond DeCoBench) and systematically emphasizes multi-stage compositional tasks rather than only atomic skills (extending beyond Colosseum). To enable structured evaluation, tasks are categorized into four types (Fig. 1(a)): **atomic (A), atomic with perturbations (AP), compositional (C), and compositional with perturbations (CP)**. This categorization disentangles different aspects of capability, allowing separate assessment of basic skill mastery, skill composition, and the robustness of both under realistic perturbations. Nevertheless, state-of-the-art Visual-Language-Action (VLA) models (Kim et al., 2024; Black et al., 2024), even when pre-trained on large-scale demonstrations, continue to struggle with composing skills in perturbed settings. Recent studies (Huang et al., 2023; Chen et al., 2025f; Black et al., 2025) have attempted to mitigate this by employing hierarchical frameworks that leverage the reasoning and planning abilities of Visual-Language Models (VLMs). However, these approaches remain fragile when combining skills under perturbations, raising a key question: *When a model fails to achieve compositional generalization in long-horizon tasks under perturbations, is the failure due to ineffective planning or insufficient execution capability?*

To this end, we introduce the second innovation of RoboHiMan, a **hierarchical evaluation paradigm** (see Fig. 1(b)) for systematically evaluating model capabilities. It includes a progressive multi-level training dataset (**L1–L4**), where L1 represents the most challenging setting with minimal atomic skill data, and L4 the easiest with larger, more diverse datasets. This progression allows analysis of how training complexity and exposure to perturbations affect long-horizon skill composition. RoboHiMan evaluates models using three modes: *Vanilla, Decoupled, and Coupled*, on a set of test tasks organized into four types shown in Fig 1(a). In Vanilla mode, the low-level policy executes tasks directly without planner guidance; Decoupled mode evaluates the planner and policy separately; and Coupled mode tests the full hierarchical system end-to-end. This setup allows

| | Diverse Perturbations | Atomic and Compositional Tasks | Progressive Multi-Level Training Dataset | Hierarchical Evaluation | Eval. of Compositional Gen. under Perturbations |
|---|---|---|---|---|---|
| RLBench (2020) | ✗ | ✗ | ✗ | ✗ | ✗ |
| CALVIN (2022) | ✗ | ✓ | ✓ | ✗ | ✗ |
| Libero-Long (2023) | ✗ | ✓ | ✓ | ✗ | ✗ |
| Colosseum (2024) | ✓ | ✗ | ✓ | ✗ | ✗ |
| VLABench (2024) | ✓ | ✓ | ✗ | ✓ | ✗ |
| DeCoBench (2025f) | ✗ | ✓ | ✗ | ✓ | ✗ |
| RoboCerebra (2025) | ✗ | ✓ | ✗ | ✓ | ✗ |
| **RoboHiMan (Ours)** | ✓ | ✓ | ✓ | ✓ | ✓ |

Table 1: **Comparison of Long-Horizon Manipulation Benchmarks.** Unlike prior benchmarks, RoboHiMan explicitly evaluates *compositional generalization under perturbations* and also covers robustness, compositionality, multi-level training, and hierarchical evaluation.

us to disentangle failures arising from planning versus execution, while systematically assessing robustness under diverse task conditions. Together with progressive training, these modes provide rich metrics to reveal detailed patterns of compositional generalization.

Through extensive experiments, we identify: **(1)** Models without a planner perform poorly when composing atomic skills, exposing the limitations of low-level policies in compositionality. **(2)** While additional training data with compositional examples improves performance, a substantial gap persists, highlighting the inherent difficulty of compositional generalization. **(3)** As shown in Fig. 1(c), VLA models such as $\pi_{0.5}$ (Black et al., 2025) perform well on LIBERO-10 (Liu et al., 2023), but fail under the diverse perturbations in HiMan-Bench, exposing limitations that prior benchmarks fail to capture. **(4)** Hierarchical systems remain brittle, as planning errors and imperfect execution compound over long horizons, leading to a sharp degradation in overall performance.

In summary, RoboHiMan makes three contributions: (1) HiMan-Bench, a novel benchmark that evaluates how well models can compose atomic skills to complete long-horizon manipulation tasks under diverse environmental perturbations. (2) A novel hierarchical evaluation paradigm that combines progressive multi-level training dataset with multiple evaluation modes, allowing separate analysis of planning and execution performance while revealing robustness limitations. (3) A comprehensive analysis of model performance, uncovering key challenges in long-horizon compositional generalization and providing insights beyond prior benchmark results.

## 2 RELATED WORKS

**Long-Horizon Robotic Manipulation Benchmarks.** Long-horizon tasks are widely regarded as a key challenge for evaluating the planning and generalization capabilities of robotic manipulation. Early benchmarks, e.g., RLBench (James et al., 2020), CALVIN (Mees et al., 2022), Libero-Long (Liu et al., 2023), and more recently RoboTwin 2.0 (Chen et al., 2025c), include such tasks but mainly train and evaluate models directly on long-horizon tasks without explicitly requiring skill composition, thus providing limited insights into task planning. Yet in practice, agents must compose learned skills to achieve long-horizon goals, beyond simple imitation (Belkhale et al., 2024; Gao et al., 2025). Recent benchmarks such as VLA-Bench (Zhang et al., 2024), RoboCasa (Nasiriany et al., 2024), DeCoBench (Chen et al., 2025f), and RoboCerebra (Han et al., 2025) introduce more challenging tasks involving language-conditioned decomposition and planning. However, they still largely treat long-horizon problems as simple skill permutations and overlook real-world perturbations (e.g., color, texture, object size) on skill composition. Colosseum (Pumacay et al., 2024) advances this line by systematically examining perturbations, revealing model vulnerabilities under environmental variations. Yet its evaluation remains mostly at the atomic-task level, where success does not guarantee robustness in compositional tasks. To this end, we propose **RoboHiMan**, which inherits Colosseum's perturbation design and further emphasizes their compounded effects in long-horizon compositional tasks. As shown in Table 1, RoboHiMan uniquely assesses *compositional generalization under perturbations*, together with robustness, skill composition, progressive training, and hierarchical evaluation.

**Vision-Language-Action Models for Long-Horizon Manipulation.** In recent years, vision–language–action (VLA) models have become a promising paradigm in robotic manipulation, processing visual and language inputs for action generation (Shao et al., 2025; Zhou et al., 2025; Ma et al., 2024). Foundation-style pretraining, as in RT-1 (Brohan et al., 2022), RT-2 (Zitkovich et al., 2023), OpenVLA (Kim et al., 2024), and $\pi_0$ (Black et al., 2024), improves generalization by

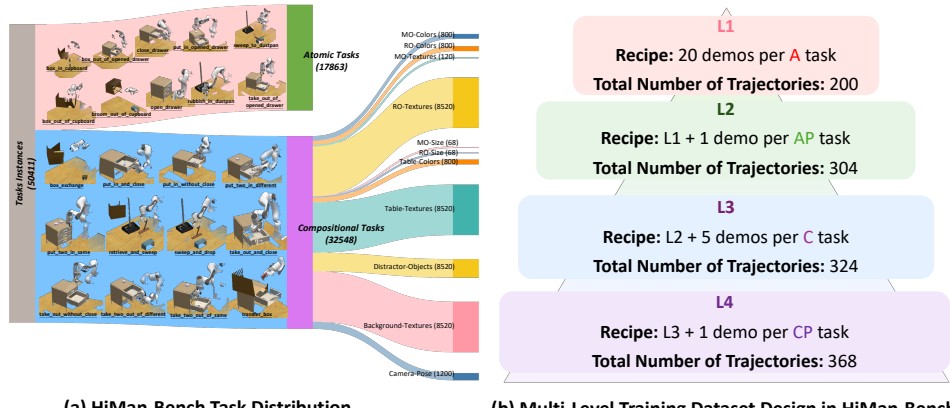

(a) HiMan-Bench Task Distribution   (b) Multi-Level Training Dataset Design in HiMan-Bench

Figure 2: This figure illustrates the key design of HiMan-Bench, including (1) **HiMan-Bench task distribution**, and (2) **multi-level training dataset design in HiMan-Bench**.

learning from large-scale trajectories. In contrast, methods such as PerAct (Shridhar et al., 2023), RVT (Goyal et al., 2023), RVT-2 (Goyal et al., 2024), and 3D Diffuser Actor (Ke et al., 2024) leverage 3D representations to achieve fine-grained action prediction. These approaches, however, often lack the explicit task-planning capabilities. To tackle long-horizon manipulation tasks, many works (Wen et al., 2024; Chen et al., 2025d; Shi et al., 2025; Wen et al., 2025; Gao et al., 2025) adopt hierarchical designs, where a foundation model decomposes instructions into sub-tasks that low-level policies execute as actions. Yet these methods face two bottlenecks: (1) reliance on complex prompt engineering and handcrafted pipelines, limiting scalability; and (2) error accumulation when low-level policies fail to reliably follow high-level plans (Han et al., 2025). In HiMan-Bench, we use natural language as the interface and evaluate both rule-based and VLM-based high-level planners paired with low-level policies (Goyal et al., 2024; Ke et al., 2024; Black et al., 2024; 2025) to systematically analyze the challenges faced at each hierarchy in complex long-horizon tasks.

## 3 ROBOHIMAN

In this section, we present **RoboHiMan**, a hierarchical evaluation paradigm for studying compositional generalization in long-horizon manipulation under perturbations. Sec. 3.1 introduces **HiMan-Bench**, a benchmark comprising both atomic and compositional tasks with diverse perturbations, along with a progressive multi-level training dataset spanning atomic to compositional skills. Sec. 3.2 outlines the **hierarchical evaluation paradigm**, which includes three modes: vanilla, decoupled, and coupled. Together, these components form a unified framework for analyzing model performance in compositional long-horizon manipulation.

### 3.1 HIMAN-BENCH

**Task and Perturbation Factors Design.** We construct **HiMan-Bench** following the task design paradigm of RLBench (James et al., 2020), implemented with the PyRep (James et al., 2019) API atop the CoppeliaSim (Rohmer et al., 2013) simulator. Building on the 10 atomic tasks and 12 compositional tasks provided by DeCoBench (Chen et al., 2025f), we leverage the Colosseum (Pumacay et al., 2024) API to extend the task set, ultimately constructing the HiMan-Bench distribution comprising 114 atomic tasks and 144 compositional tasks. Each atomic task consists of two sub-stages, segmented by discrete robot-state changes to capture fundamental manipulator-object interactions (James & Davison, 2022; Chen et al., 2025f;d), which are further composed into multi-stage tasks. Some tasks require cross-domain transfer (e.g., from drawer to cupboard manipulation). Even with atomic skills mastered, models must still correctly schedule and compose them to follow long-horizon language instructions such as "*take the strawberry jello out of the drawer and put it into the cupboard*", highlighting challenges in long-horizon planning, cross-domain generalization, and robust skill composition.

Specifically, to systematically investigate the role of skill composition in long-horizon tasks and its robustness under environmental perturbations, we adopt 12 perturbation factors introduced in Colos-

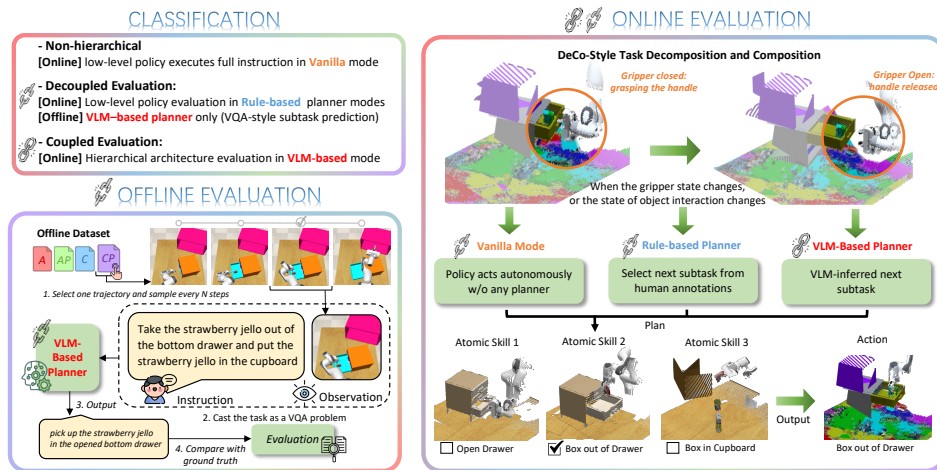

**Figure 3: The overview of hierarchical evaluation paradigm**

seum (Pumacay et al., 2024): manipulation object color (`MO_Color`), texture (`MO_Texture`), and size (`MO_Size`); receiver object color (`RO_Color`), texture (`RO_Texture`), and size (`RO_Size`); light color (`Light_Color`); table color (`Table_Color`) and texture (`Table_Texture`); distractor objects (`Distractor`); background texture (`Background_Texture`); and camera pose (`Camera_Pose`). The perturbation space covers 20 colors, 213 textures, and 78 distractor objects sampled from the YCB Object Dataset (Çalli et al., 2015). Object size scaling ranges depend on the specific task (e.g., cupboard $[0.75, 1.15]$, drawer $[0.9, 1.15]$). Lighting perturbations are applied by sampling RGB values from $[0.0, 0.0, 0.0]$ to $[0.5, 0.5, 0.5]$, while camera perturbations are applied to three viewpoints (front, left_shoulder, right_shoulder) with position offsets in $[-0.1, -0.1, -0.1]$ to $[0.1, 0.1, 0.1]$ and Euler-angle perturbations in $[-0.05, -0.05, -0.05]$ to $[0.05, 0.05, 0.05]$. This design aligns with existing benchmarks while extending evaluation to more challenging tasks, thereby enabling systematic assessment of robustness and generalization. Fig. 2(a) illustrates the distribution of atomic and compositional task instances across different perturbation factors and variants in HiMan-Bench. **For clarity, HiMan-Bench organizes tasks into four categories: atomic (A)-10 tasks, atomic with perturbations (AP)-104 tasks, compositional (C)-12 tasks, and compositional with perturbations (CP)-132 tasks.** Additional implementation details are provided in Appendix B.

**Multi-level Training Dataset Design.** HiMan-Bench proposes a hierarchical training data design to systematically investigate the impact of different data "recipes" on generalization performance. This design covers configurations ranging from the most challenging to the most comprehensive, strictly following a progressive order from difficult to easy (*from difficult to easy*). The construction details of each layer (data recipes and the number of expert demonstrations) are summarized in Fig. 2(b). **L1**: Contains demonstrations of A tasks, with 20 demonstrations for each task. This is the most challenging setting. If a model trained on this dataset performs well on compositional tasks, it indicates strong compositional generalization ability. **L2**: Builds upon L1 by introducing AP tasks, with 1 demonstration per AP task. The goal is to improve robustness across diverse variants, which is crucial for reducing error accumulation in long-horizon tasks. **L3**: Extends L2 by including demonstrations of 4 C tasks ( *put_in_without_close*, *sweep_and_drop*, *take_out_without_close*, and *transfer_box*), with 5 demonstrations per task. This allows the model to directly observe part of the multi-step compositional processes. **L4**: Further extends L3 by introducing CP tasks for the 4 C tasks, with 1 demonstration per CP task. This exposes the model to more compositional scenarios.

## 3.2 HIERARCHICAL EVALUATION PARADIGM

RoboHiMan employs a hierarchical evaluation paradigm for different models: a high-level planner first decomposes the instruction into subtasks relevant to the current stage, while a low-level policy executes these subtasks by generating robot actions. Formally, both the planner and the policy take as input a natural language instruction $l$ and a visual observation $o$, where $o$ can be either multi-view 2D signals (e.g., RGB images) or 3D representations (e.g., point clouds). The planner then produces a subtask description $s$, which the low-level policy translates into a sequence of robot actions $\{a_{1:T}\}$.

For comparison, we also consider a non-hierarchical baseline, in which the planner is omitted and the low-level policy directly maps $(l, o)$ to $\{a_{1:T}\}$. This contrast enables explicit evaluation of the contribution of hierarchical design to generalization in long-horizon compositional tasks.

**Evaluation Paradigm.** As illustrated in Fig. 3, the RoboHiMan evaluation paradigm is organized into three settings: **1) Vanilla (Non-hierarchical, Online).** The low-level policy executes the entire task online directly from the original instruction without planner, serving as a non-hierarchical baseline. **2) Decoupled (Hierarchical, Planner and Policy Evaluated Separately).** This paradigm aims to analyze the limitations of the high-level planner and low-level policy independently, and includes two variants: *i) Rule-based Planner (Online)*: A rule-based planner schedules subtasks online based on robot state changes, with transition boundaries given by annotations. Following De-CoBench (Chen et al., 2025f), we use physical interaction changes between the gripper and objects to determine transitions. Despite its heuristic nature, this mode provides a strong baseline for the low-level policy. *ii) VLM-based Planner (Offline)*: A vision-language model is evaluated as the planner in an offline setting. The model predicts the current subtask at fixed intervals, and its planning accuracy is measured in a VQA-style evaluation, reflecting its ability in scene understanding and task decomposition. **3) Coupled (Hierarchical, Online).** The full hierarchical architecture is deployed online. The VLM-based planner generates subtask descriptions upon detecting gripper state transitions, which are then executed by the low-level policy. This setting evaluates the end-to-end integration of planning and execution.

# 4 EXPERIMENTS

We conduct experiments to address the questions below: **Q1: Skill Composition Without Planning (Sec. 4.2).** Can models without a planner reliably combine atomic skills to solve tasks? **Q2: Scaling Effects of Training Data (Sec. 4.3).** How does performance change as training data expands from atomic to compositional tasks with perturbations? **Q3: Sensitivity to Perturbations (Sec. 4.4).** Which perturbations most hinder skill composition, and how do models handle them? **Q4: Generalization to Unseen Compositions (Sec. 4.5).** Can models generalize to new task compositions beyond those seen in training? **Q5: Bottlenecks in Hierarchical Architectures (Sec. 4.6).** Do failures stem from flawed planning, weak execution, or both? **Q6: Real-World Validation (Sec. 4.7).** Do real-world tasks face similar challenges, and can hierarchical architectures help?

## 4.1 EXPERIMENTAL SETUP

All simulation experiments are conducted on the proposed HiMan-Bench tasks introduced in Sec. 3.1, while the detailed setup of the real-world experiments is provided in the Appendix D.

**High-Level Planner.** We adopt Qwen2.5-VL (Bai et al., 2025) as the vision-language model (VLM) backbone for the high-level planner. The training process is based on frames sampled at fixed intervals from demonstration data, using the current frame's visual observation and the full task instruction as input, and the corresponding subtask description as output. The model is fine-tuned on the HiMan-Bench dataset, with training and inference prompts detailed in Appendix C.1.

**Low-Level Policy.** We select four state-of-the-art VLA models (RVT-2 (Goyal et al., 2024), 3D Diffuser Actor (Ke et al., 2024), $\pi_0$ (Black et al., 2024), and $\pi_{0.5}$ (Black et al., 2025)) as low-level policies. For each baseline model, we follow the original policy design but modify the handling of language inputs. Since language plays a crucial role in distinguishing stages and guiding skill composition, we provide stage-specific instructions at different execution points. The input-output formats of each baseline model are summarized in Table 3, and the implementation details are provided in Appendix C.1.

**Evaluation Metric.** For atomic tasks, we generate 720 episodes for evaluation, and for compositional tasks, 900 episodes. For each task, we include 15 episodes without perturbations (denoted as None) and 5 episodes for each perturbation factor, plus 5 episodes with all perturbations enabled (denoted as All). During online evaluation, the environment is configured to match these test episodes. Because of variations in object placement or workspace sampling, some offline episodes are not fully reproducible online. We therefore report results only on valid episodes, which account for about 90% of the total. Performance is reported as the average success rate over atomic tasks (A), perturbed atomic tasks (AP), compositional tasks (C), and perturbed compositional tasks (CP). For

offline evaluation, we measure only the high-level planner's subtask prediction accuracy. Frames are sampled every 10 steps for atomic tasks and every 30 steps for compositional tasks, and each sampled frame is modeled as a VQA instance to evaluate planning accuracy.

## 4.2 SKILL COMPOSITION WITHOUT PLANNING

To address **Q1**, we train four baseline VLA models on HiMan-Bench's multi-level datasets and evaluate them on diverse test sets. In Fig. 1(c), models without a planner (Vanilla) show marginal gains from richer data, and their performance on both compositional (C) and perturbed compositional (CP) tasks remains near zero, indicating that models without a planner cannot compose atomic skills into coherent long-horizon behaviors. In contrast, rule-based planner (RP) variants achieve substantial improvements, especially with compositional (L3) and perturbed (L4) training data.

> **Finding:** *(i) Data diversity and scale offer limited benefits for Vanilla models, slightly improving robustness but failing to enable skill composition. (ii) Explicit planning is essential, as it supports robust skill composition and underscores the role of hierarchical reasoning in complex long-horizon tasks.*

## 4.3 SCALING EFFECTS OF TRAINING DATA

Regarding **Q2**, the results in Fig. 4 illustrate the scaling effects of multi-level training data under a rule-based planner. For evaluations on atomic tasks (A&AP), all models show an upward trend in performance from L1 to L2. This indicates that adding expert trajectories of atomic skills under perturbations can indeed enhance model robustness. However, after in-

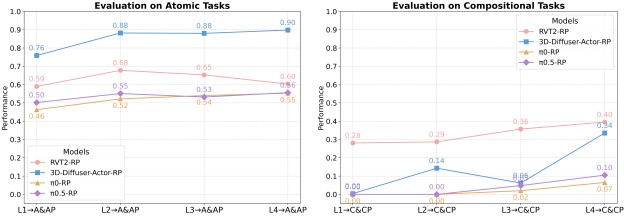

Figure 4: Performance scaling curves on atomic and compositional tasks under different scaling levels.

corporating compositional skill data, the performance on atomic tasks improves only marginally and may even degrade in some cases. In contrast, for all compositional tasks (C&CP), models trained only with atomic-level data (L1, L2) fail to generalize to compositional settings. Although $\pi_0$-RP, $\pi_{0.5}$-RP, and RVT2-RP show some improvement when trained with L3 and L4 data, the gains remain marginal. By comparison, 3D-Diffuser-Actor-RP benefits from L2 training with modest generalization gains, but its performance drops at L3 and improves again at L4. For compositional tasks, even with multi-level data including perturbations and compositional demonstrations, the generalization ability of current models remains highly constrained, revealing a clear bottleneck.

> **Finding:** *(i) Scaling atomic-skill data improves atomic performance and, as robust atomic execution is a prerequisite, also benefits compositional tasks. (ii) Increasing both the quantity and diversity of compositional-skill training data further enhances model performance on compositional tasks. However, the overall success rate remains low, even though all compositional tasks can in principle be solved by combining the atomic skills the model has learned.*

## 4.4 SENSITIVITY OF SKILL COMPOSITION TO PERTURBATIONS.

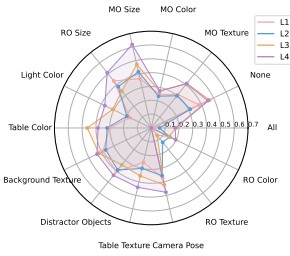

(a) Perturbation effects across training data levels.

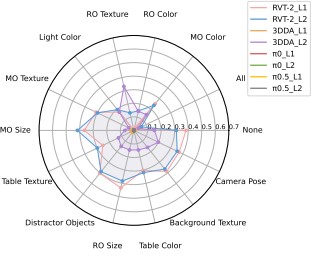

(b) Comparison of models at L1–L2.

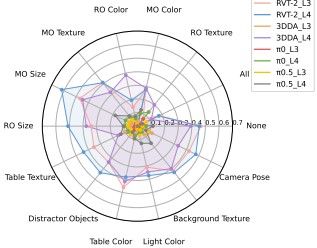

(c) Comparison of models at L3–L4.

Figure 5: Robustness under perturbations across different settings.

To investigate **Q3**, we conduct robustness experiments to systematically analyze the effects of perturbations across different tasks and model scales. As shown in Fig. 5, model robustness exhibits consistent trends under various perturbation factors. Fig. 5(a) shows that skill composition is particularly sensitive to perturbations such as object color (`MO_Color`, `RO_Color`), texture (`MO_Texture`, `RO_Texture`), and all factors enabled (`All`). Models trained only on atomic data (L1, L2) display limited adaptability, whereas introducing even a small amount of perturbed data in compositional tasks (L3, L4) leads to improvements: the model not only learns skill composition but also becomes more robust to unseen variations in appearance, geometry, and viewpoint. Fig. 5(b) and Fig. 5(c) further compare different architectures. RVT-2 and 3D Diffuser-Actor consistently outperform the baseline policies $\pi_0$ and $\pi_{0.5}$ across all training scales, while $\pi_{0.5}$ performs better than $\pi_0$.

> ***Finding:*** *(i) For compositional tasks with various perturbations, including perturbed compositional-skill data in training, effectively improves the model's robustness in performing compositional tasks. In comparison, adding perturbed atomic-skill data alone provides only limited gains in robustness. (ii) Both data design and architectural inductive biases (e.g., keyframe selection, 3D information integration) contribute to improved generalization and robustness.*

## 4.5 Compositional Generalization to Unseen Tasks

To answer **Q4**, we evaluate the baseline VLA models after L4 training, where testing covers 12 compositional tasks and their perturbed versions (C&CP), among which 4 tasks were already seen in training (see Sec. 3.1 and Appendix C for details). As shown in Fig. 6, RVT2-RP and 3D Diffuser-Actor-RP achieve relatively low success rates even on the seen C&CP tasks, indicating that the models have not sufficiently mastered the corresponding compositional skills. On the unseen tasks, the success rates remain similarly limited, further suggesting a lack of effective compositional generalization. In contrast, while $\pi_0$ and $\pi_{0.5}$ achieve moderate performance on seen tasks, they almost completely fail on unseen tasks, which further highlights their lack of generalization.

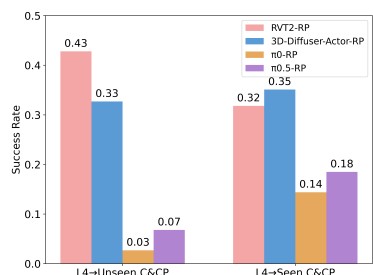

Figure 6: Generalization performance on seen/unseen compositional tasks.

> ***Finding:*** *Even with partial exposure to compositional skills during training, current models still show clear limitations in learning and utilizing skill compositions, making it difficult to achieve true compositional generalization on unseen tasks.*

## 4.6 Bottlenecks in Hierarchical Architectures

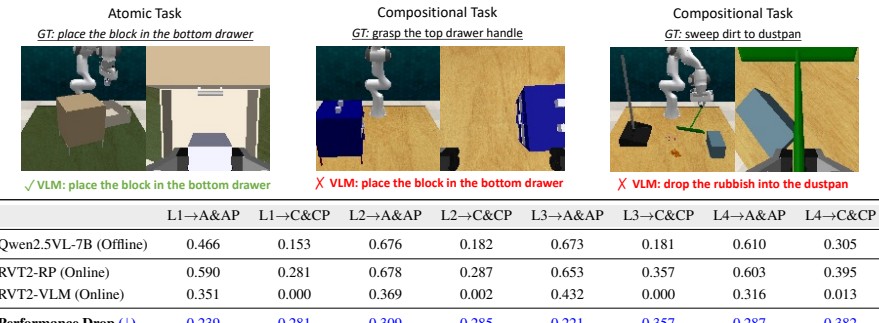

| | L1→A&AP | L1→C&CP | L2→A&AP | L2→C&CP | L3→A&AP | L3→C&CP | L4→A&AP | L4→C&CP |
|---|---|---|---|---|---|---|---|---|
| Qwen2.5VL-7B (Offline) | 0.466 | 0.153 | 0.676 | 0.182 | 0.673 | 0.181 | 0.610 | 0.305 |
| RVT2-RP (Online) | 0.590 | 0.281 | 0.678 | 0.287 | 0.653 | 0.357 | 0.603 | 0.395 |
| RVT2-VLM (Online) | 0.351 | 0.000 | 0.369 | 0.002 | 0.432 | 0.000 | 0.316 | 0.013 |
| **Performance Drop** (↓) | 0.239 | 0.281 | 0.309 | 0.285 | 0.221 | 0.357 | 0.287 | 0.382 |

Table 2: **Comparison of offline vs. online evaluation.** Blue numbers show RVT2-VLM performance drops relative to RVT2-RP.

Regarding **Q5**, Table 2 reveals the core bottlenecks of hierarchical architectures. In offline evaluation, when the planner is trained solely on atomic skill data, its generalization to unseen compositional skills is clearly limited; even after introducing some compositional skills during training, the success rate improves but remains relatively low. In online evaluation, this issue is further amplified, and the impact of planning errors becomes more pronounced. Specifically, in online evaluation, both RVT2-RP and RVT2-VLM maintain strong performance on atomic tasks and their perturbed variants

(A&AP). However, on compositional tasks and perturbed compositional tasks (C&CP), RVT2-VLM shows a significant performance drop compared to RVT2-RP. In particular, in L4→C&CP setting, its success rate **decreases by 0.382**, revealing marked vulnerability. Further analysis indicates that performance degradation in complex tasks stems from the compounded effects of planning failures and low-level policy execution issues. The VLM-based high-level planner fails to fully leverage the capabilities of the low-level policy when dealing with long-horizon compositional tasks.

> **Finding:** *The bottlenecks of hierarchical architectures stem from three main issues: (i) the high-level planner may generate incorrect plans. (ii) the low-level policy may fail during execution. (iii) if the hierarchical system is not properly designed, failures at the high or low level are not effectively handled, leading to error accumulation and eventual task failure.*

## 4.7 REAL-WORLD VALIDATION

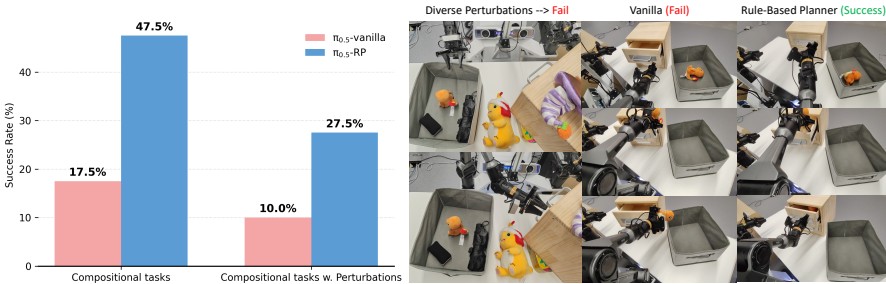

Figure 7: **Real World Experimental Results.**

To answer **Q6**, we conduct real-world validation experiments (Appendix D). We designed a small set of atomic skills, their long-horizon compositions, and diverse perturbations (e.g., distractors, object position changes, and human interventions). In Fig. 7, end-to-end execution without a planner ($\pi_{0.5}$-vanilla) achieved only **17.5%** success on compositional tasks, dropping further to **10.0%** under perturbations. In contrast, pairing the same low-level policy with a rule-based planner(where the arm moves to the next human-annotated sub-instruction whenever it stays idle near the initial pose for a fixed duration), $\pi_{0.5}$-RP substantially improved performance to **47.5%** and **27.5%**, respectively. **These results confirm: (i) real-world long-horizon manipulation indeed faces compositional generalization challenges under perturbations, and (ii) hierarchical architectures demonstrate clear potential benefits, achieving higher performance when an idealized planner selects sub-instructions.**

## 5 CONCLUSION

In this work, we propose RoboHiMan, a hierarchical evaluation paradigm for studying *compositional generalization under perturbations* in long-horizon manipulation. RoboHiMan first introduces a novel benchmark, HiMan-Bench, which evaluates the ability of different VLA models to compose atomic skills into long-horizon behaviors under diverse perturbations. In addition, we design three evaluation paradigms that can effectively disentangle the sources of planning and execution failures across progressively expanded training settings. Based on extensive experiments, we draw the following key conclusions: (1) Compositional skill learning is intrinsically challenging, and simply scaling up data cannot fundamentally solve this problem; (2) Model robustness to perturbations is as critical as compositionality itself; (3) Hierarchical systems require stronger feedback mechanisms to achieve effective coordination between planning and execution.

Looking forward, RoboHiMan opens up promising research directions for the robotics and VLA communities. Future work should explore **robust skill composition mechanisms**, **feedback-rich hierarchical architectures**, and **perturbation-aware training recipe** that improve robustness under distribution shifts. Equally important is the development of **scalable compositional datasets** and **new evaluation metrics** that go beyond task success to capture error recovery and skill reusability. By providing both a challenging benchmark and an analytical framework, RoboHiMan aims to accelerate progress toward building generalizable robotic agents capable of reliable long-horizon manipulation in realistic environments.

ETHICS STATEMENT

Our experiments are limited to desktop-level robot manipulation in simulated and controlled environments. As such, we do not expect our work to pose significant societal risks. Future work should consider safety when extending to real-world scenarios.

REPRODUCIBILITY STATEMENT

We have made resources to facilitate reproduction of our results publicly accessible. Specifically, our anonymous project repository (https://robohiman.github.io/ ) provides code, documentation, and example visualizations of our experiments. Detailed experimental settings can be found in Appendix C.

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

## A  THE USE OF LARGE LANGUAGE MODELS

We used a large language model (LLM) as a general-purpose tool to assist with writing and polishing the manuscript; all ideas, experiments, and analyses are our own, and we take full responsibility for the content.

## B  TASK DESIGN IN SIMULATION

HiMan-Bench consists of 114 atomic tasks (A&AP) shown in Fig. 8 and 144 compositional tasks (C&CP) shown in Fig. 9. The specific task types without perturbations follow the same design as DecoBench (Chen et al., 2025f), while the perturbation types are selected from Colosseum (Pumacay et al., 2024). For completeness, we provide a detailed description of the task specifications here.

### B.1  ATOMIC TASKS

**(1) open_drawer**
*Description:* Grasp the <top/middle/bottom> drawer handle; Pull the <top/middle/bottom> drawer open.
*Success Metric:* Success when the specified drawer is opened by at least 0.15 meters.

**(2) close_drawer**
*Description:* Move close to the <top/middle/bottom> drawer handle; Push the <top/middle/bottom> drawer shut.
*Success Metric:* Success when the target drawer is pushed to within 0.03 meters of its fully closed position.

**(3) put_in_opened_drawer**
*Description:* Pick up the block on the drawer's surface; Place the block in the <top/middle/bottom> drawer.
*Success Metric:* Success when the block is detected by the proximity sensor inside the target drawer.

**(4) take_out_of_opened_drawer**
*Description:* Pick up the block in the <top/middle/bottom> drawer; Place the block on the drawer's surface.
*Success Metric:* Success when the block is detected by the proximity sensor on the drawer's surface.

**(5) box_out_of_opened_drawer**
*Description:* Pick up the strawberry jello box in the <top/middle/bottom> drawer; Place it on the drawer's surface.
*Success Metric:* Success when the jello box is detected on the drawer's surface, outside of the drawer.

**(6) box_in_cupboard**
*Description:* Pick up the <strawberry jello/spam/sugar> on the table; Place the item in the cupboard.
*Success Metric:* Success when the item is detected inside the cupboard by a proximity sensor.

**(7) box_out_of_cupboard**
*Description:* Pick up the <strawberry jello/spam/sugar> in the cupboard; Place the item on the table.
*Success Metric:* Success when the item is detected on the table by a proximity sensor.

**(8) broom_out_of_cupboard**
*Description:* Pick up the broom in the cupboard; Place the broom on the table.
*Success Metric:* Success when the broom is detected on the table by a proximity sensor.

**(9) sweep_to_dustpan**
*Description:* Pick up the broom on the table; Sweep dirt into the dustpan.
*Success Metric:* Success when all dirt particles are detected in the dustpan by a proximity sensor.

**(10) rubbish_in_dustpan**
*Description:* Pick up the rubbish on the table; Drop the rubbish into the dustpan.
*Success Metric:* Success when the rubbish is detected inside the dustpan by a proximity sensor.

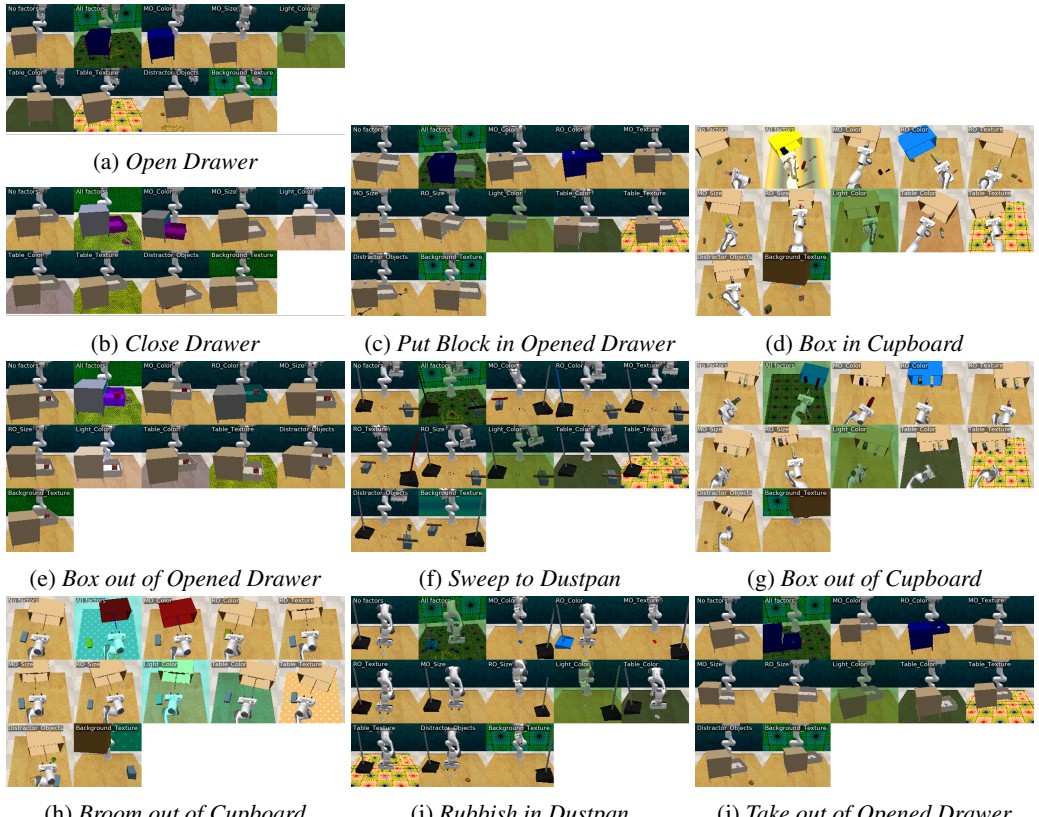

(a) *Open Drawer*

(b) *Close Drawer*      (c) *Put Block in Opened Drawer*      (d) *Box in Cupboard*

(e) *Box out of Opened Drawer*      (f) *Sweep to Dustpan*      (g) *Box out of Cupboard*

(h) *Broom out of Cupboard*      (i) *Rubbish in Dustpan*      (j) *Take out of Opened Drawer*

Figure 8: Atomic tasks with perturbations.

## B.2 COMPOSITIONAL TASKS

**(1) put_in_without_close**
*Description:* Grasp the <top/middle/bottom> drawer handle; Pull the <top/middle/bottom> drawer open; Pick up the block on the drawer's surface; Place the block in the <top/middle/bottom> drawer.
*Success Metric:* Success when the block is detected inside the specified drawer by a proximity sensor.

**(2) take_out_without_close**
*Description:* Grasp the <top/middle/bottom> drawer handle; Pull the <top/middle/bottom> drawer open; Pick up the block in the <top/middle/bottom> drawer; Place the block on the drawer's surface.
*Success Metric:* Success when the block is detected on the drawer's surface by a proximity sensor.

**(3) put_in_and_close**
*Description:* Grasp the <top/middle/bottom> drawer handle; Pull the <top/middle/bottom> drawer open; Pick up the block on the drawer's surface; Place the block in the <top/middle/bottom> drawer; Push the <top/middle/bottom> drawer shut.
*Success Metric:* Success when the block is inside the specified drawer and the drawer is closed.

**(4) take_out_and_close**
*Description:* Grasp the <top/middle/bottom> drawer handle; Pull the <top/middle/bottom> drawer open; Pick up the block in the <top/middle/bottom> drawer; Place the block on the drawer's

surface; Push the <top/middle/bottom> drawer shut.
*Success Metric:* Success when the block is on the drawer's surface and the drawer is closed.

**(5) put_two_in_same**
*Description:* Grasp the <top/middle/bottom> drawer handle; Pull the drawer open; Pick up the first block on the drawer's surface; Place the first block in the drawer; Pick up the second block on the drawer's surface; Place the second block in the same drawer.
*Success Metric:* Success when both blocks are detected inside the specified drawer.

**(6) take_two_out_of_same**
*Description:* Grasp the <top/middle/bottom> drawer handle; Pull the drawer open; Pick up the first block in the drawer; Place the first block on the drawer's surface; Pick up the second block in the drawer; Place the second block on the drawer's surface.
*Success Metric:* Success when both blocks are detected on the drawer's surface.

**(7) put_two_in_different**
*Description:* Grasp the first drawer handle; Pull the drawer open; Pick up the first block on the drawer's surface; Place the first block in the first drawer; Push the drawer shut; Grasp the second drawer handle; Pull the drawer open; Pick up the second block; Place the second block in the second drawer.
*Success Metric:* Success when each block is detected inside its corresponding drawer.

**(8) take_two_out_of_different**
*Description:* Grasp the first drawer handle; Pull the drawer open; Pick up the first block in the drawer; Place the first block on the drawer's surface; Push the drawer shut; Grasp the second drawer handle; Pull the drawer open; Pick up the second block; Place the second block on the drawer's surface.
*Success Metric:* Success when both blocks are detected on the drawer's surface.

**(9) box_exchange**
*Description:* Pick up the sugar in the cupboard; Place the sugar on the table; Pick up the spam on the table; Place the spam in the cupboard.
*Success Metric:* Success when the sugar is on the table and the spam is in the cupboard.

**(10) sweep_and_drop**
*Description:* Pick up the rubbish on the table; Drop the rubbish into the dustpan; Pick up the broom; Sweep dirt into the dustpan.
*Success Metric:* Success when all dirt pieces and rubbish are detected in the dustpan.

**(11) transfer_box**
*Description:* Grasp the <top/middle/bottom> drawer handle; Pull the drawer open; Pick up the strawberry jello in the drawer; Place the strawberry jello in the cupboard.
*Success Metric:* Success when the strawberry jello is detected inside the cupboard.

**(12) retrieve_and_sweep**
*Description:* Pick up the broom in the cupboard; Sweep dirt into the dustpan.
*Success Metric:* Success when all dirt pieces are detected in the dustpan.

# C  ADDITIONAL EXPERIMENTS IN SIMULATION

## C.1  IMPLEMENTATION DETAILS

**Low-level Policy.** We select four state-of-the-art VLA models (RVT-2 (Goyal et al., 2024), 3D Diffuser Actor (Ke et al., 2024), $\pi_0$ (Black et al., 2024), and $\pi_{0.5}$ (Black et al., 2025)) as low-level policies. **RVT-2**: A two-stage multi-view transformer that predicts coarse regions of interest and refines gripper poses using zoomed-in views. Trained with 4 views, batch size 24, for 15 epochs (100k steps). **3D Diffuser Actor**: A conditional 3D diffusion transformer that integrates tokenized 3D scene representations, language, and proprioception. Trained with a batch size of 8 for 600k steps. $\pi_0$: A VLA transformer that combines a pre-trained VLM with a continuous-action expert via flow matching. Trained with a batch size of 32 for 50k steps, with an action chunk size of 50. $\pi_{0.5}$: Extends $\pi_0$ by incorporating multimodal inputs and hierarchical inference, enabling broader generalization. Trained with the same batch size and settings as $\pi_0$.

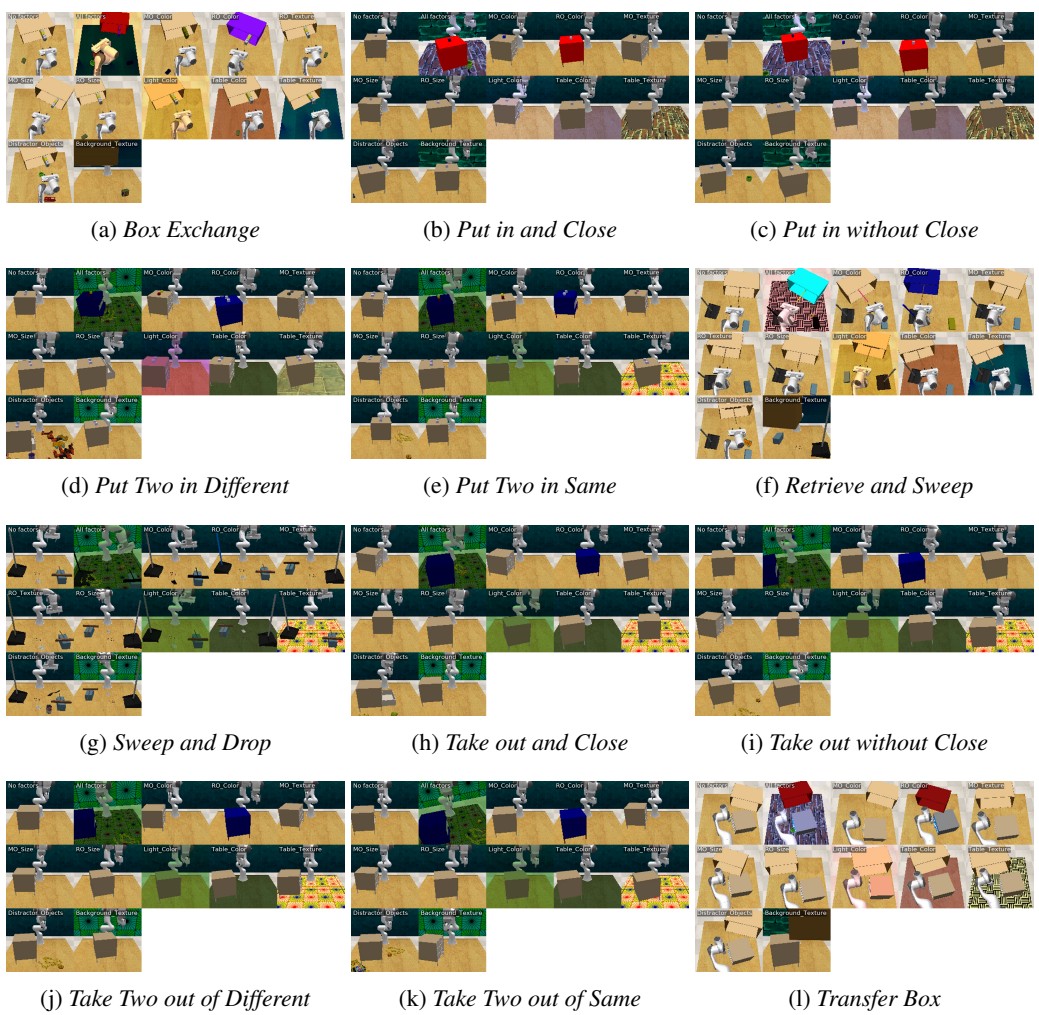

(a) *Box Exchange*        (b) *Put in and Close*        (c) *Put in without Close*

(d) *Put Two in Different*    (e) *Put Two in Same*    (f) *Retrieve and Sweep*

(g) *Sweep and Drop*    (h) *Take out and Close*    (i) *Take out without Close*

(j) *Take Two out of Different*    (k) *Take Two out of Same*    (l) *Transfer Box*

Figure 9: Compositional tasks with perturbations.

Table 3 summarizes the configuration of the low-level policies used in our hierarchical framework. RVT-2 and 3D Diffuser Actor take multi-view RGB images from front, wrist, and left/right shoulder cameras, while $\pi_0$ and $\pi_{0.5}$ receive RGB images from front and wrist cameras only. For language processing, RVT-2 and 3D Diffuser Actor use CLIP (Radford et al., 2021), whereas $\pi_0$ and $\pi_{0.5}$ use PaliGemma-3B (Beyer et al., 2024). Action prediction is modeled differently for these groups: RVT-2 and 3D Diffuser Actor follow the paradigm used in (James & Davison, 2022; James et al., 2022; Shridhar et al., 2023), predicting the next keypoint in the trajectory rather than the full trajectory, which reduces learning difficulty and improves training efficiency. In contrast, $\pi_0$ and $\pi_{0.5}$ are trained on data sampled at 20 Hz, with an action horizon of 50 frames.

| Model | View | Vision Modality | Language Encoder | Action |
|---|---|---|---|---|
| RVT-2 | Front & Wrist & Left Shoulder & Right Shoulder | Multi-View Re-rendered RGB | CLIP | Next Keypoint Prediction & EEF Pose |
| 3D Diffuser Actor | Front & Wrist & Left Shoulder & Right Shoulder | RGB-D | CLIP | Next Keypoint Prediction & EEF Pose |
| $\pi 0$ | Front & Wrist | RGB | PaliGemma-3B | Trajectory & Joint |
| $\pi 0.5$ | Front & Wrist | RGB | PaliGemma-3B | Trajectory & Joint |

Table 3: Input, output, and language encoder configurations of low-level policies.

**High-level planner.** We use Qwen2.5-VL (Bai et al., 2025) as an implementation of a high-level planner. To construct training data, we sample one frame every ten frames from the robot trajectories as input, and the output includes the reasoning and the current sub-task stage. The template for predicting the next sub-task is shown in Table 4. For generating reasoning data, we use the prompt template in Table 5, where we provide the sampled frame, the sub-task description, and the frame at which the sub-task ends, and the model generates reasoning conditioned on the answer, i.e., explaining why the sub-task should be performed based only on the start frame and task instruction.

---

**Prompt Template: Next Sub-task Prediction**

This is a tabletop manipulation scene with a Franka robotic arm. The task is: {`task_description`}.

You are given {`image_paths|length`} different views of the same scene: {`image_paths`}

Based on the task and the current scene, determine what the robot should do next (the next sub-task).

Output in the following XML format: `<reasoning>` natural language description of reasoning `</reasoning>` `<sub_task>` natural language description of the next sub-task `</sub_task>`

---

Table 4: Prompt Template for Next Sub-task Prediction in VLM-based Planner.

---

**Prompt Template: Sub-task Reasoning**

This is a tabletop manipulation scene with a Franka robotic arm. The task is: {`task_description`}.

You are given the following images:

- Start frame views: {`start_frame_images`}

- End frame views [just for understanding]: {`end_frame_images`}

The sub-task performed to move from the start frame towards the end frame is: {`sub_task`}

Provide reasoning for why this sub-task should be performed next, based only on the start frame and task instruction. Do not use information from the end frame in your reasoning.

Output in the following XML format: `<reasoning>` Describe the scene first, then explain why this sub-task is chosen based on the start frame and task instruction only. `</reasoning>`

---

Table 5: Prompt Template for Sub-task Reasoning in VLM-based Planner.

## C.2 DETAILED RESULTS

This section presents detailed experimental results. Table 6 summarizes the performance of different hierarchical frameworks trained with varying levels of data across different types of tasks. Table 7 lists the variation factors corresponding to the numeric headers used in the following tables. Tables 8–11 report the performance of each model under different perturbations for both atomic and compositional tasks, providing a comprehensive view of how various models handle disturbances in the environment.

## D EXPERIMENTAL SETUP FOR REAL WORLD

**Robot Setup.** We use *Cobot Mobile ALOHA*, a robot based on the Mobile ALOHA system design (Fu et al., 2025). It is equipped with two wrist cameras and a front camera. In our experiments, we primarily use the right arm, which provides 6-DoF joints plus one gripper degree of freedom, while the left arm remains static.

| | L1 → A&AP | L1 → C&CP | L2 → A&AP | L2 → C&CP | L3 → A&AP | L3 → C&CP | L4 → A&AP | L4 → C&CP |
|---|---|---|---|---|---|---|---|---|
| RVT2-RP | 0.590 | 0.281 | 0.678 | 0.287 | 0.653 | 0.357 | 0.603 | 0.395 |
| 3D-Diffuser-Actor-RP | 0.759 | 0.004 | 0.882 | 0.143 | 0.880 | 0.062 | 0.898 | 0.335 |
| $\pi_0$-RP | 0.463 | 0 | 0.522 | 0 | 0.540 | 0.020 | 0.553 | 0.065 |
| $\pi_{0.5}$-RP | 0.502 | 0 | 0.551 | 0 | 0.532 | 0.048 | 0.556 | 0.105 |
| RVT2-Vanilla | 0.092 | 0 | 0.105 | 0.001 | 0.063 | 0.001 | 0.117 | 0 |
| 3D-Diffuser-Actor-Vanilla | 0.103 | 0 | 0.235 | 0 | 0.052 | 0 | 0.259 | 0 |
| $\pi_0$-Vanilla | 0.153 | 0 | 0.168 | 0 | 0.126 | 0 | 0.162 | 0.005 |
| $\pi_{0.5}$-Vanilla | 0.085 | 0 | 0.089 | 0 | 0.091 | 0 | 0.099 | 0.006 |

Table 6: Performance across different training levels (L1–L4) and test task (A&AP, C&CP).

| Index | 0 | 1 | 2 | 3 | 4 | 5 | 6 |
|---|---|---|---|---|---|---|---|
| Perturbation | No variation factors | All variation factors | MO_Color | RO_Color | MO_Texture | RO_Texture | MO_Size |
| Index | 7 | 8 | 9 | 10 | 11 | 12 | 14 |
| Perturbation | RO_Size | Light_Color | Table_Color | Table_Texture | Distractor_Objects | Background_Texture | Camera_Pose |

Table 7: Description of variation factor indices.

| | 0 | 1 | 2 | 3 | 4 | 5 | 6 | 7 | 8 | 9 | 10 | 11 | 12 | 14 |
|---|---|---|---|---|---|---|---|---|---|---|---|---|---|---|
| L1 → A&AP | 0.707 | 0.087 | 0.391 | 0.385 | 0.611 | 0.480 | 0.800 | 0.641 | 0.511 | 0.578 | 0.729 | 0.681 | 0.739 | 0.660 |
| L1 → C&CP | 0.384 | 0.018 | 0.250 | 0.017 | 0.310 | 0 | 0.366 | 0.435 | 0.172 | 0.302 | 0.255 | 0.404 | 0.388 | 0.370 |
| L2 → A&AP | 0.771 | 0.326 | 0.543 | 0.447 | 0.722 | 0.680 | 0.711 | 0.795 | 0.783 | 0.756 | 0.604 | 0.652 | 0.826 | 0.681 |
| L2 → C&CP | 0.313 | 0.060 | 0.236 | 0.148 | 0.298 | 0.133 | 0.417 | 0.383 | 0.196 | 0.319 | 0.300 | 0.389 | 0.365 | 0.354 |
| L3 → A&AP | 0.757 | 0.227 | 0.467 | 0.605 | 0.778 | 0.520 | 0.744 | 0.692 | 0.622 | 0.667 | 0.688 | 0.638 | 0.783 | 0.745 |
| L3 → C&CP | 0.467 | 0.179 | 0.273 | 0.143 | 0.412 | 0.067 | 0.471 | 0.341 | 0.310 | 0.462 | 0.356 | 0.341 | 0.421 | 0.422 |
| L4 → A&AP | 0.686 | 0.318 | 0.578 | 0.526 | 0.611 | 0.360 | 0.625 | 0.641 | 0.600 | 0.622 | 0.688 | 0.644 | 0.609 | 0.638 |
| L4 → C&CP | 0.455 | 0.174 | 0.275 | 0.196 | 0.409 | 0 | 0.619 | 0.511 | 0.373 | 0.385 | 0.440 | 0.438 | 0.435 | 0.477 |

Table 8: RVT2-RP performance across different training levels, test tasks, and perturbations.

| | 0 | 1 | 2 | 3 | 4 | 5 | 6 | 7 | 8 | 9 | 10 | 11 | 12 | 14 |
|---|---|---|---|---|---|---|---|---|---|---|---|---|---|---|
| L1 → A&AP | 0.946 | 0.051 | 0.750 | 0.811 | 0.800 | 0.920 | 0.812 | 0.875 | 0.795 | 0.936 | 0.159 | 0.727 | 0.773 | 0.957 |
| L1 → C&CP | 0.006 | 0 | 0 | 0 | 0 | 0 | 0.023 | 0 | 0 | 0 | 0 | 0.019 | 0 | 0 |
| L2 → A&AP | 0.151 | 0.037 | 0.148 | 0.145 | 0.043 | 0.333 | 0.067 | 0.148 | 0.173 | 0.148 | 0.127 | 0.151 | 0.161 | 0.200 |
| L2 → C&CP | 0.971 | 0.452 | 0.957 | 0.944 | 0.882 | 0.840 | 0.821 | 0.882 | 0.957 | 0.896 | 0.702 | 0.891 | 0.913 | 1.000 |
| L3 → A&AP | 0.955 | 0.333 | 0.978 | 0.973 | 0.938 | 0.920 | 0.806 | 0.969 | 0.978 | 0.979 | 0.711 | 0.909 | 0.750 | 0.978 |
| L3 → C&CP | 0.063 | 0.041 | 0.037 | 0.077 | 0 | 0 | 0.050 | 0.055 | 0.020 | 0.098 | 0.104 | 0.078 | 0.080 | 0.120 |
| L4 → A&AP | 0.955 | 0.512 | 0.935 | 0.946 | 0.875 | 0.960 | 0.861 | 0.938 | 0.978 | 0.979 | 0.689 | 0.932 | 0.909 | 0.978 |
| L4 → C&CP | 0.393 | 0.113 | 0.286 | 0.385 | 0.319 | 0.200 | 0.447 | 0.377 | 0.346 | 0.418 | 0.157 | 0.339 | 0.400 | 0.333 |

Table 9: 3D-Diffuser-Actor-RP performance across different training levels, test tasks, and perturbations.

| | 0 | 1 | 2 | 3 | 4 | 5 | 6 | 7 | 8 | 9 | 10 | 11 | 12 | 14 |
|---|---|---|---|---|---|---|---|---|---|---|---|---|---|---|
| L1 → A&AP | 0.468 | 0.300 | 0.409 | 0.579 | 0.444 | 0.320 | 0.488 | 0.441 | 0.465 | 0.545 | 0.545 | 0.391 | 0.556 | 0.442 |
| L1 → C&CP | 0 | 0 | 0 | 0 | 0 | 0 | 0 | 0 | 0 | 0 | 0 | 0 | 0 | 0 |
| L2 → A&AP | 0.582 | 0.275 | 0.500 | 0.658 | 0.500 | 0.400 | 0.512 | 0.500 | 0.442 | 0.581 | 0.545 | 0.478 | 0.622 | 0.512 |
| L2 → C&CP | 0 | 0 | 0 | 0 | 0 | 0 | 0 | 0 | 0 | 0 | 0 | 0 | 0 | 0 |
| L3 → A&AP | 0.574 | 0.289 | 0.533 | 0.711 | 0.556 | 0.360 | 0.585 | 0.500 | 0.488 | 0.545 | 0.636 | 0.578 | 0.533 | 0.512 |
| L3 → C&CP | 0.024 | 0.017 | 0 | 0.018 | 0.021 | 0.067 | 0 | 0.018 | 0.057 | 0 | 0.034 | 0.036 | 0 | 0.017 |
| L4 → A&AP | 0.560 | 0.300 | 0.578 | 0.658 | 0.722 | 0.440 | 0.605 | 0.588 | 0.628 | 0.581 | 0.568 | 0.511 | 0.600 | 0.442 |
| L4 → C&CP | 0.063 | 0 | 0.096 | 0.057 | 0.087 | 0.133 | 0.098 | 0.089 | 0.056 | 0.071 | 0.077 | 0.056 | 0.054 | 0.036 |

Table 10: $\pi_0$-RP performance across different training levels, test tasks, and perturbations.

**Task Design.** For the real-world setting, we design a set of atomic and compositional tasks to verify whether the challenges highlighted in RoboHiMan also arise in physical environments. The

| | 0 | 1 | 2 | 3 | 4 | 5 | 6 | 7 | 8 | 9 | 10 | 11 | 12 | 14 |
|---|---|---|---|---|---|---|---|---|---|---|---|---|---|---|
| L1 → A&AP | 0.503 | 0.310 | 0.511 | 0.553 | 0.611 | 0.560 | 0.600 | 0.514 | 0.455 | 0.511 | 0.556 | 0.391 | 0.596 | 0.455 |
| L1 → C&CP | 0 | 0 | 0 | 0 | 0 | 0 | 0 | 0 | 0 | 0 | 0 | 0 | 0 | 0 |
| L2 → A&AP | 0.566 | 0.366 | 0.543 | 0.553 | 0.778 | 0.480 | 0.500 | 0.657 | 0.545 | 0.545 | 0.667 | 0.404 | 0.574 | 0.614 |
| L2 → C&CP | 0 | 0 | 0 | 0 | 0 | 0 | 0 | 0 | 0 | 0 | 0 | 0 | 0 | 0 |
| L3 → A&AP | 0.524 | 0.238 | 0.574 | 0.605 | 0.722 | 0.238 | 0.524 | 0.657 | 0.568 | 0.614 | 0.578 | 0.478 | 0.617 | 0.500 |
| L3 → C&CP | 0.037 | 0.074 | 0 | 0.019 | 0.060 | 0 | 0.070 | 0.070 | 0.069 | 0.018 | 0.051 | 0.075 | 0.036 | 0.088 |
| L4 → A&AP | 0.569 | 0.238 | 0.587 | 0.632 | 0.667 | 0.440 | 0.524 | 0.629 | 0.659 | 0.545 | 0.533 | 0.543 | 0.681 | 0.523 |
| L4 → C&CP | 0.114 | 0.040 | 0.123 | 0.074 | 0.096 | 0 | 0.182 | 0.103 | 0.130 | 0.093 | 0.136 | 0.057 | 0.161 | 0.069 |

Table 11: $\pi_{0.5}$-RP performance across different training levels, test tasks, and perturbations.

atomic tasks include four skills as shown in Fig. 10: (1) open the top drawer (open_drawer), (2) close the top drawer (close_drawer), (3) pick an item from the box and place it on the table (box_item_on_table), and (4) pick an item from the table and place it into the opened top drawer (table_item_in_opened_drawer). On top of these primitives, we compose four long-horizon tasks as shown in Fig. 11: (1) pick an item from the table, put it on the opened drawer, and then close it (table_item_in_opened_drawer_close), (2) open the top drawer, pick an item from the table, put it inside, and then close the drawer (table_item_in_drawer), (3) pick an item from the box, put it into the opened drawer, and then close it (box_item_in_opened_drawer_close), and (4) open the top drawer, pick an item from the box, put it into the drawer, and then close the drawer (box_item_in_drawer).

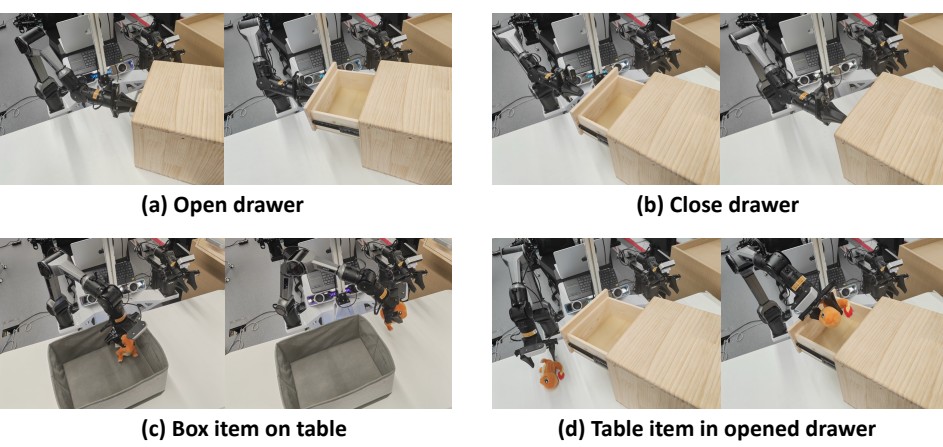

**(a) Open drawer**  **(b) Close drawer**

**(c) Box item on table**  **(d) Table item in opened drawer**

Figure 10: **Atomic Tasks in Real World.**

To assess robustness, we further introduce three perturbation factors: (i) distractors, by placing irrelevant objects in the scene; (ii) positional variations, by changing the initial locations or heights of objects; and (iii) human interventions, such as moving objects during execution or occluding the camera. These tasks and perturbations are not intended as a complete real-world benchmark, but rather as evidence that the issues identified in RoboHiMan are also encountered in real physical settings.

**Training data.** For training, we collected demonstrations covering both atomic and compositional tasks. Specifically, each of the four atomic tasks—open the top drawer (open_drawer), close the top drawer (close_drawer), pick an item from the box and place it on the table (box_item_on_table), and pick an item from the table and place it into the opened top drawer (table_item_in_opened_drawer)—was recorded with 40 demonstrations each. For compositional tasks, we only selected one long-horizon task, namely opening the top drawer, picking an item from the box, putting it into the drawer, and then closing the drawer (box_item_in_drawer), for which we recorded a single demonstration. In addition, for all selected tasks (both atomic and compositional), we recorded one extra demonstration under each of the perturbation settings, including distractors, object position variations, and human interventions. This setup ensures that the

training data not only covers core skills but also explicitly exposes the model to diverse real-world perturbations.

**Evaluation.** For real-world experiments, we adopted $\pi_{0.5}$ as the low-level policy. Two evaluation modes were considered: (i) directly executing the original instruction without any planner, and (ii) using a rule-based planner, similar to the simulation setup, to decompose the task into subgoals. For each compositional task, we evaluated 20 episodes in total, consisting of 10 trials without perturbations and 10 trials with perturbations. The average success rate across these episodes was reported as the final performance metric.

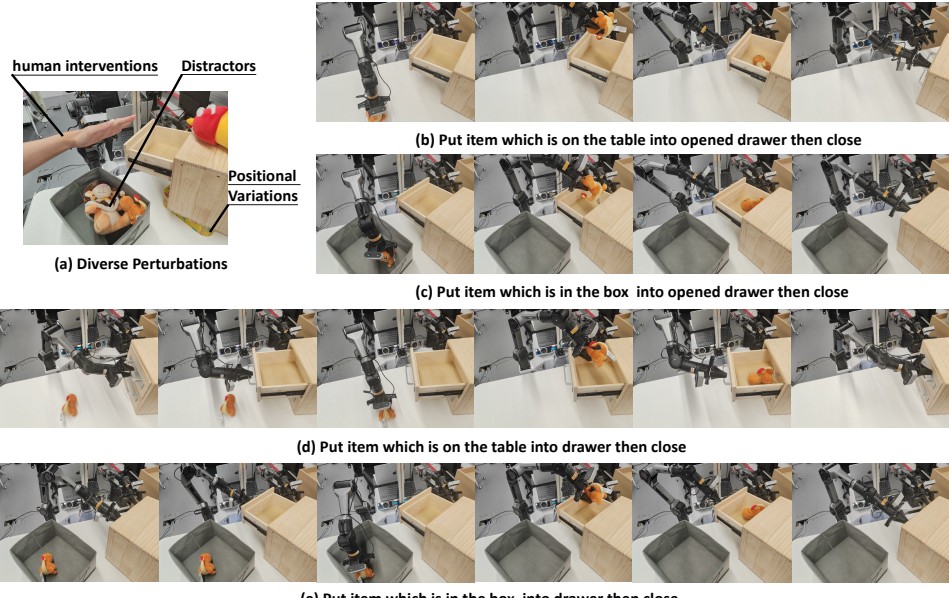

Figure 11: **Compositional Tasks in Real World.**

