# OpenReview forum: "RoboHiMan: A Hierarchical Evaluation Paradigm for Compositional Generalization in Long-Horizon Manipulation"
_ICLR.cc/2026/Conference — ICLR 2026 Conference Withdrawn Submission_

### Official Review · Reviewer_yffQ · 2025-10-27

**Soundness:** 3
**Presentation:** 4
**Contribution:** 3
**Rating:** 4
**Confidence:** 3

**Summary:**

Studying compositional generalization for sequential decision-making tasks is very timely and impactful. The paper proposes a new benchmark and experiment design choices for evaluating the requirement of hierarchical policies and when they fail for long-horizon manipulation. First, they show that the considered tasks indeed require task planning along with low-level planners, and then study particular failure cases based on the type of data the low-level policy is trained on and the type of high-level planner: rule-based or a VLM. The findings make sense: they validate the importance and failure cases of learning based hierarchical approaches for solving long-horizon tasks and how the performance depends on the characteristics of training data.

**Strengths:**

1. The paper performs an extensive evaluation using one VLM high-level planner (Qwen-2.5) and four VLA low-level policies. Real world results are also shown to support findings.

2. The benchmark introduces a large set of compositional tasks and atomic skills.

**Weaknesses:**

1. The training details are very unclear. Is it training from scratch or finetuning based setup? The amount of training data is provide for real-world, what about the simulation experiments?

2. I am also concerned about the clarity of the evaluation pipeline. It will be great if simple pseudocodes or flow charts can be provided to explain the hierarchical evaluation mechanism.

3. The main innovation of the paper is the finding that VLA's by themselves are not sufficient to solve long-horizon manipulation tasks even with aligned training data. While VLMs are a great alternative to a rule-based planner, they often fail to understand the context of the problem. Either the former fails, or the latter or both leading to low success rates. The study is quite limited to a single task planner and four low-level VLA policies (authors claim SOTA without reference), compromising the diversity of the study and the findings. It would be great to see MolmoAct, OpenVLA(-OFT) or even Octo.

**Questions:**

1. DecoBench already provides the atomic tasks and compositional tasks and Colosseum benchmark is all about perturbations, is the main novelty here just a blend of both?

2. How is the whole study of high-level planner failure and low-level policy failure methodologically different from established practices in TAMP? For example, failure of hierarchical systems due to compounded planning errors is quite prevalent in TAMP, thus requiring integrated task and motion planning-based methods. Finally I am not sure about this, but I feel like the authors should also have some learning for TAMP (like T2M https://arxiv.org/abs/2303.12153) algorithms in the table since the whole like of research on TAMP is about solving the hierarchical symbolic-geometric long-horizon planning.

3. How do you train the same VLA policy for both atomic and compositional training data?

---

### Official Review · Reviewer_6d5t · 2025-10-31

**Soundness:** 2
**Presentation:** 3
**Contribution:** 1
**Rating:** 2
**Confidence:** 3

**Summary:**

This paper introduces RoboHiMan, a benchmark for studying visual and compositional generalization. The paper first outlines the multi-level hierarchy of the benchmark and then provides an experimental evaluation on the suggested tasks as well as various ablations.

**Strengths:**

**Clarity**
* The paper is well written and structure. The language is clear and easy to follow.

**Related Work**
* The related work correctly outlines other work that has been drawn from.

**Benchmark purpose**
* In general, a benchmark should satisfy a specific purpose with well defined metrics that one ought to measure. This work provides task layouts and metrics to measure in this sense.

**Experimental evaluation**
* The experiment setup seems quite extensive.

**Weaknesses:**

**Clarity**
* The plots are very difficult to read. For example to see the results in Figure 1, I had to zoom so far that the plot would almost cover my whole screen.
* After reading the paper, I still have no idea what the tasks look like. It is not clear to me how tasks compose and when or why I would expect compositional generalization. That is likely because the tasks simply stem from a different benchmark and were not designed in this paper. However, a benchmark paper should make it clear what the tasks look like.
* Relatedly, the observation information is not described properly. For example, the resolution of the images, the structure of the language command and how they are compositionally related is not described.
* It is unclear to me what section 4.7 contributes to the benchmark as this setup is not something someone else might be able to use to run their experiments.

**Novelty**
* The paper seems to largely combine two existing benchmarks that already study a form of compositionality and visual generalization.

**Related Work**
* The related work solely focuses on very recent work but the idea of controlling robots from visual observation has been around for a while. There are only 3 papers in the whole references that refer to papers older than 2020.

**Motivation**
* The motivation for the work is a bit unclear to me. A benchmark is in general designed to study specific properties, such as visual generalization and composition. This benchmark now combines these two properties just to then claim that they can easily be disentangled again in this benchmark. That seems a bit circular to me.

**Empirical Design, Claims and Evidence**
* None of the experiments have any variance indicators limiting our ability to make our statistical significance of the results.
* Many of the claims are overstatements with great generality that are not supported by evidence. For example: “Data diversity and scale offer limited benefits for Vanilla models, slightly improving robustness but failing to enable skill composition” - This is a very broad statement. All we can say is that on the provided tasks adding a single trajectory per task of the more complex chain is insufficient. When people talk about scale they are not talking about 100 trajectories. Large pretraining datasets usually have 70k trajectories and more. The phrasing of this finding makes it sound like that it is not useful to collect this much data which is clearly not true from prior work.
* While the benchmark's goal is to disentangle the failure modes of visual and compositional generalization, the results seem to just entangle them. For example in section 4.4 and 4.5, the paper states that compositional tasks benefit more from visual perturbations than atomic ones. Is it unclear to me whether this is because more longer horizon trajectories were added which makes composition easier or because more trajectories with different colors which makes visual reasoning easier.

**Questions:**

None

---

### Official Review · Reviewer_K56r · 2025-10-31

**Soundness:** 2
**Presentation:** 3
**Contribution:** 2
**Rating:** 4
**Confidence:** 3

**Summary:**

The paper studies the problem of compositional generalization in long-horizon tasks. It introduces a benchmark with a hierarchical evaluation paradigm that enables systematic analysis of existing models’ generalization capabilities.

**Strengths:**

The study tackles a fundamental problem in robot learning --- generalization across long-horizon tasks. The proposed public benchmark provides a valuable testbed for fair comparison and detailed analysis of learned models. The paper also includes real-world experiments that further validate the benchmark’s utility.

**Weaknesses:**

The set of evaluated methods is relatively limited --- only four VLA models are compared, and a single VLM model is used as the high-level planner. A broader combination of high-level planning approaches and low-level control policies should be explored to strengthen the conclusions.

The reported results appear somewhat inconsistent and noisy. For instance, in Fig. 4, the model’s performance on certain dimensions (e.g., receiver object color and table color) decreases when trained with additional data augmentation --- the L4 configuration underperforms compared to L2 and L3.

Moreover, the paper lacks discussion of several related benchmarks and empirical studies that also focus on compositional generalization [1, 2]. Incorporating these discussions would help contextualize the proposed benchmark and clarify its contributions.


[1] What Matters in Learning from Large-Scale Datasets for Robot Manipulation, ICLR’25;

[2] Efficient Data Collection for Robotic Manipulation via Compositional Generalization, RSS’24.

**Questions:**

What could be the underlying reasons for the observed fluctuations in model performance? Increasing the diversity of training data does not appear to consistently improve results, at least for certain factors. Providing further analysis or investigation into these variations would help clarify the model’s behavior and strengthen the overall study.

Additionally, the evaluations are somewhat limited --- more extensive comparisons with alternative models and learning frameworks would be valuable to better support the conclusions and demonstrate the robustness of the proposed benchmark.

---

### Official Review · Reviewer_U74C · 2025-11-02

**Soundness:** 2
**Presentation:** 3
**Contribution:** 2
**Rating:** 4
**Confidence:** 4

**Summary:**

This paper proposes a benchmark for hierarchical long horizon robot manipulation to evaluate the capabilities of different models. While prior works have focused on creating benchmarks that focus on observation perturbations (e.g. object sizes, colors etc), these works mostly focus on short horizon skills (atomic skills) or a simple chaining of few such skills. By contrast, one of the main contributions of the current paper is a compositional generalization benchmark. Experiments are performed using 4 policies including 3D diffuser actor, pi_0.5 and shows that simply training on compositional tasks does not bring compositional generalization.

**Strengths:**

The paper is well written and focuses on an important problem. Compositional generalization in robot manipulation is a long standing problem and while VLMs do offer many advantages in solving it they still sometimes fall short. Hence, having a benchmark to quantify progress in this domain will be extremely useful to the community.

**Weaknesses:**

The paper focuses on too much on results and very little into how the benchmark construction was guided. Looking at the compositional tasks in the appendix, all of them seem pretty straight forward and small variations of atomic tasks. For instance, there are compositional tasks such as put in a block without closing the drawer or taking out from an open drawer. While there are atomic tasks such as put in open drawer, or open drawer. These are not very challenging tasks from a task orchestration perspective. I think the real challenge lies in handling low-level policy failures or multi-view images or low-image resolutions. However, none of these challenges are discussed or considered. The paper mostly just focuses on atomic skills and compositional skills and simply training on the data together.  I think even a simple strategy in which the model is asked to output the stage of the task that it thinks it currently in, before performing the task could be a strong baseline.

Overall, I think the paper is tackling an extremely important problem statement. However, in it’s current state it is missing many crucial evaluations and insights. If we can add these the paper would make for a strong contribution.

**Questions:**

What are the results for the offline planning metric? How well does the high level planning model perform in this setting?

What if we use a more structured approach for the VLM in which we ask it to locate obects (using bounding boxes), identify object states (open or close),  maybe identify spatial relations and then ask the model to reason about what sub-task to take. I am pretty sure such a structured/guided approach would work much better than simply asking the model to output the subtask.

*History*: What happens when we use history for the planner? Clearly, telling the model what tasks/instructions it has tried performing and their state could be helpful for the model.

Also, given the compositional tasks are not really long horizon but simply 4-5 skills being performed one after the other, what happens in a truly long context task. For example, giving 6 different colored cups and each cup needs to be put into a container or something similar.

How are the images fed to the planning model? RLBench has all multi-view images at each step (4-5), how are they being used by the model, also what image resolution is being used? I think for many tasks the objects can be too small from front pixels.

---

### Note · Authors · 2025-12-10

I have read and agree with the venue's withdrawal policy on behalf of myself and my co-authors.